# Metabolite profiling, antifungal, biofilm formation prevention and disruption of mature biofilm activities of *Erythrina senegalensis* stem bark extract against *Candida albicans and Candida glabrata*

**Benjamin Kingsley Harley**[1]*, **Anthony Martin Quagraine**[1], **David Neglo**[2], **Mike Okweesi Aggrey**[1], **Emmanuel Orman**[3], **Nana Ama Mireku-Gyimah**[4], **Cedric Dzidzor Amengor**[3], **Jonathan Jato**[1], **Yussif Saaka**[5], **Theophilus Christian Fleischer**[1]

1 Department of Pharmacognosy and Herbal Medicine, School of Pharmacy, University of Health and Allied Sciences, Ho, Ghana, 2 Department of Basic Science, School of Basic and Biomedical Sciences, University of Health and Allied Sciences, Ho, Ghana, 3 Department of Pharmaceutical Chemistry, School of Pharmacy, University of Health and Allied Sciences, Ho, Ghana, 4 Department of Pharmacognosy and Herbal Medicine, School of Pharmacy, University of Ghana, Accra, Ghana, 5 Department of Pharmaceutics, School of Pharmacy, University of Health and Allied Sciences, Ho, Ghana

* bkharley@uhas.edu.gh

## Abstract

The antifungal activity of the 70% ethanol stem bark extract of *Erythrina senegalensis* (ESB) against different strains and drug resistant clinical isolates of *Candida albicans* and *Candida glabrata* were evaluated in the study. The effect of ESB on biofilms as well as its activity in combination with fluconazole, nystatin or caspofungin against the *Candida* strains were also evaluated. We then evaluated the antifungal activity of a microemulsion formulation of ESB against planktonic and biofilms of the *Candida* species. UPLC-QTOF-MS$^2$ analysis was then undertaken to identify the phytoconstituents of the extract and UPLC fingerprints developed for the routine authentication as part of quality control measures. ESB exerted strong antifungal activities against *C. albicans* ATCC 10231 and SC5314 strains, and *C. glabrata* ATCC 2001 strain with minimum inhibitory concentration (MIC) values from 3.91 to 31.25 µg/mL and minimum fungicidal concentrations (MFCs) that ranged from 62.5 to 250 µg/mL. It also exhibited potent antifungal activities (MIC = 4–64 µg/mL) against a collection of *C. albicans* and *C. glabrata* clinical isolates that were resistant to either nystatin or azole antifungals. The formulated ESB demonstrated higher antifungal potency against the *C. albicans* and *C. glabrata* strains with MIC values of 3.91–31.25 µg/mL which was the same as the MFC values. The extract and its microemulsion formulation were active against biofilms of the strains of the *Candida* species inhibiting their biofilm formations (SMIC$_{50}$ = 16–64 µg/mL) and their preformed biofilms (SMIC$_{50}$ = 128 −>512 µg/mL). ESB also exhibited synergistic antifungal action with fluconazole and nystatin against *C. albicans* ATCC 10231 and *C. glabrata* ATCC 2001 strains in the checkerboard assay. Chemical characterization of the extract revealed the presence of phenolic compounds such as flavonoids and their prenylated derivatives, anthracene glycosides and alkaloids. UPLC Fingerprints of the

**Data Availability Statement:** All relevant data are within the paper and its Supporting Information files.

**Funding:** The author(s) received no specific funding for this work.

**Competing interests:** The authors have declared that no competing interests exist.

extract was also developed and validated for routine identification and authentication of the stem bark of *E. senegalensis*. The study findings have demonstrated that the stem bark of *E. senegalensis* is as a potential source of bioactive compounds that could be developed as novel antifungal agents.

## 1. Introduction

Vulvovaginal candidiasis (VVC) is the most common infection affecting between 70–75% of women at least once in their lifetime, and it is estimated that about half will experience a recurrence [1]. The infection can be triggered by various factors such as antibiotic therapy, immunosuppressive therapy, hormonal fluctuations during pregnancy, hormone replacement therapy and metabolic disorders such as diabetes and stress [2]. Even though *C. albicans* is the most common VVC causative agent, the identification of non-albicans *Candida* (NAC) species, mainly *C. glabrata*, as the causative pathogen of this infection appears to be continuously increasing [3]. The ability of Candida species to infect diverse hosts as a result of a wide range of virulent factors such as their morphological transition between yeast and hyphal forms, expression of adhesins and invasins on cell surfaces, thigmotropism, phenotypic switching, secretion of hydrolytic enzymes and formation of biofilms. The biofilms formed are intrinsically resistant to antifungal drugs at therapeutic concentrations effective against the non-adhering cells, thus needing higher concentrations resulting in a plethora of adverse effects such as hepatotoxicity and nephrotoxicity [4].

Currently used antifungal drugs in the treatment of VVC are grouped into three major classes: azoles, polyenes and echinocandins [5]. The fungistatic azoles (e.g., voriconazole and fluconazole) primarily inhibit cytochrome P450-dependent conversion of lanosterol to ergosterol by inhibiting the demethylase enzyme Erg11 from the ergosterol biosynthesis pathway [6]. The polyenes (e.g., nystatin) disrupts fungal cell membrane via binding to ergosterol in the fungal cell membrane and are fungicidal against *C. albicans* whiles echinocandins (e.g., caspofungin), the most recently developed class of antifungal agents, are fungicidal against *C. albicans* by inhibiting the synthesis of the cell wall crosslinking component β-1,3-glucan [7]. Although new derivatives within these classes have been introduced over the years, novel classes of antifungals have not [8]. The limited antifungal arsenal both in terms of classes and the number of agents within each class present several clinical problems. As stated earlier, these antifungal agents have reduced efficacy against *Candida* biofilms compared to planktonic cells. Furthermore, extensive use and overexposure of these antifungals particularly the azoles have resulted in resistance in the *Candida* species [9]. Thus, there is an urgent call for the discovery and development of new antifungal agents.

*Erythrina senegalensis* DC, a prickly medium-sized shrub that grows up to 5–15 m high belongs to the family Fabaceae. It is a plant native to West tropical Africa and grows from Senegal to Northern Cameroon [10]. Its trifoliate leaves which are lanceolate to broadly ovate and glabrous are variable in size with the central leaflet being largest [11]. *E. senegalensis* also has very rough and fissured bark which becomes very conspicuous as the plant ages. Although the seeds are poisonous, the plant is used extensively in traditional medicine [12]. The stem bark is used in the treatment of throat inflammation, bronchitis, jaundice, yellow fever, malaria, and sexually transmitted infections [13]. Pounded bark and leaves of *E. senegalensis* are given to pregnant women during childbirth and after delivery to ease the pain. The stem bark and the roots are also employed in the treatment of haemorrhoids, leprosy, and gastrointestinal

disorders [14]. Chemical investigations of *E. senegalensis* has led to the isolation of several prenylated isoflavonoids such as derrone, auriculatin, alpinumisoflavone, 8-prenylluteone, 6,8-diprenylgenistein, erysenegalensein D, erysenegalensein N, erysenegalensein O [15] and epoxyisoflavones like erysenegalensein F and erysenegalensein G [16]. The presence of some of these flavonoids has also been confirmed through LC-MS analysis of extracts of the stem bark [17]. Some of the isolated compounds have been reported to possess anti-HIV [18], anti-cancer [19], antimicrobial [13], and antidiabetic [20] activities justifying some of its traditional uses. Extracts of the plant have also been shown to demonstrate antimalaria, antihypertensive, antidiabetic, hepatoprotective and antioxidant activities [21,22].

Despite its numerous medicinal uses and biological investigations, *E. senegalensis* has rarely been evaluated for its antifungal activity. There is a critical need for new treatments for *Candida* infections considering that susceptibility patterns are changing globally with the emergence of resistant *Candida* strains [23]. We, therefore, sought to evaluate the antifungal activity of the hydroethanolic stem bark extract of *E. senegalensis* against different strains of *C. albicans* and *C. glabrata*, the two most common causative pathogens of VVC. Furthermore, we identified the constituents of the extract by employing Reversed-Phase Ultra-High-Performance Liquid Chromatography (RP-UPLC) coupled to Quadrupole-Time-Of-Flight (QTOF) Tandem Mass Spectrometer (MS).

## 2. Materials and methods

### 2.1 Chemicals and reagents

Sabouraud Dextrose Agar (SDA), Mueller-Hinton (MH) agar, RPMI 1640 media, XTT assay reagent, voriconazole and chloramphenicol were obtained from Thermo Fisher (Oxoid Limited, Hampshire, UK). Other drugs used in the experiment include fluconazole (Pfizer Inc., New York, NY, United States), nystatin and caspofungin (Sigma Chemical Co., St. Louis, MO, United States).

### 2.2 Collection, processing and extraction of *E. senegalensis* stem bark

The stem bark of *E. senegalensis* was collected from Ejura, Ashanti Region of Ghana in May 2020. They were authenticated by Mr. Alfred Ofori at the Institute of Traditional and Alternative Medicine (ITAM), University of Health and Allied Sciences (UHAS) where voucher specimens were deposited (Voucher number: UHAS/ITAM/2020/SB006). The collected materials were cleaned thoroughly, chopped into pieces, air-dried for 7 days, and afterwards pulverised into coarse powder.

An amount of 720 g each of the powdered material was Soxhlet-extracted using 70% ethanol to mimic the traditional methods of preparation of the plant material. After 24 h, the extract obtained was evaporated on a rotary evaporator at 50˚C [24]. The extract (ESB) weighing 46.15 g (Yield: 6.41%) was then kept in a desiccator until needed for use.

### 2.3 Formulation of a microemulsion of *E. senegalensis* stem bark extract

The microemulsion of ESB was prepared by incorporating the extract in a nano-system consisting of Phosphate Buffer Saline (PBS) (pH 7.4) as aqueous phase, 10% cholesterol as oil phase and surfactant mixture made of soybean phosphatidylcholine (SPC)/10% polyoxyethylene (20) cetyl ether as per the protocol of Bonifácio *et al*., 2015 [25]. The formulation was obtained by sonicating in a rod sonicator (Q700 –Qsonica® 700 W) with an ice bath for 10 min with 30 s intervals every 2 min. After preparing the formulation, ESB was then added to the mixture and the sonication process was performed again for 3 min in continuous manner.

## 2.4 Antifungal testing

**2.4.1 Fungal strains and isolates.** *C. albicans* ATCC 10231 and SC5314 strains, and *C. glabrata* ATCC 2001 strain were bought from Thermo Scientific™ (Waltham, MA USA).

Clinical isolates of *C. albicans* and *C. glabrata* harvested from pregnant women with VVC were collected from the Microbiology Department, Ho Teaching Hospital, Ghana. Discrete fungal colonies were sub-cultured on SDA augmented with chloramphenicol before incubation for 2 days at 37˚C to ensure pure *Candida* isolates were obtained. Isolates of *C. albicans* and *C. glabrata* were identified by culturing the pure *Candida* isolates on HiCrome Candida Differential Agar (HiMedia Laboratories, India) at 35˚C for 2 days for production of species-specific colours. *C. albicans* isolates appeared as light green coloured smooth colonies whereas *C. glabrata* isolates appeared as glistening cream-coloured convex and smooth colonies. They were selected and used for the study [26].

**2.4.2 Antifungal susceptibility testing of the *Candida* isolates.** The susceptibility of the *Candida* isolates against fluconazole (25 μg), voriconazole (1 μg) or nystatin (100 μg) was investigated by the disk diffusion method on MH agar containing 2% glucose and 0.5 μg/ml methylene blue with slight modifications [27]. Using a sterile inoculating loop, an inoculum of distinct colonies of *Candida* isolates from the SDA plates were transferred into 5 mL test tubes containing 0.85% sterile saline solution and emulsified to form a suspension of turbidity equivalent to 0.5 McFarland standard as compared to 0.5 McF PhoenixSpec Calibrator (Becton, Dickinson and Company, USA). Thereafter, media lawns were seeded in three dimensions using sterile swabs which were dipped into the prepared inoculum. Afterward, antifungal-loaded disks were placed aseptically on the lawn and incubated at 37˚C for 24 to 48 h. The zone diameters of antifungal disks were measured using a ruler.

For fluconazole, zone diameters of $\geq 19$ mm were considered susceptible, 15–18 mm dose-dependently sensitive, and $\leq 14$ mm considered resistant. For voriconazole, zone diameters of $\geq 17$ mm were considered susceptible, 14–16 mm dose-dependently sensitive, and $\leq 13$ mm considered resistant. For nystatin, zone diameters of $\geq 25$ mm were considered susceptible, 17–24 mm dose-dependently sensitive, and $\leq 16$ mm considered resistant [26].

**2.4.3 Evaluation of antifungal activity.** The antifungal activity of the 70% ethanol *E. senegalensis* stem bark extract (ESB)(free) and when it is formulated into a microemulsion against *C. albicans* ATCC 10231 and SC5314 strains and *C. glabrata* ATCC 2001 strain was evaluated using the microbroth dilution technique based on document M27—A3 by the Clinical and Laboratory Standards Institute (CLSI) (Clinical and Laboratory Standards Institute, 2008) with slight modifications [28]. Briefly, a 4 mg/mL stock solution of each test ample was prepared in 2% DMSO. Two-fold serial dilutions of these stocks was prepared until 10 different concentrations were obtained. An aliquot of 100 μL of double strength Mueller Hinton broth was dispensed into wells of 96-well plates and mixed with 100 μL of the test samples to obtain well concentrations ranging from 1000–1.95 μg/mL. Wells 11 and 12 served as microorganism control (Broth + organism only) and negative control (Broth with no organism) respectively. As positive controls, voriconazole and nystatin at 128–0.125 μg/mL were evaluated in separate plates against the *C. albicans* strains. 100 μL of 0.5 McFarland standardized, test organisms were then added to the wells and the plates incubated at 37˚C for 48 hours. The MIC values were the evaluated visually and spectrophotometrically at 490 nm after the addition of 10 μL of 1.0% 2,3,5-triphenyltetrazolium chloride developer (Reatec®) to each well followed by incubation of the microplates at 37˚C for 20 min.

In addition, the free extract was also evaluated against several *C. albicans* and *C. glabrata* clinical isolates with different levels of resistance against azoles or echinocandins. Fluconazole and caspofungin were employed as positive controls.

The antifungal activities were interpreted as follows: very strong bioactivity <3.515 μg/mL; strong bioactivity 3.515–25 μg/mL; moderate bioactivity 26–100 μg/mL; weak bioactivity 101–500 μg/mL; very weak bioactivity 501–2000 μg/mL; and no activity >2000 μg/mL [29].

**2.4.4 Determination of Minimum Fungicidal Concentration (MFC).** To evaluate the fungicidal effect of the test samples against the *Candida* strains, aliquots from each well from the antifungal susceptibility assay were transferred onto SDA plates and incubated at 37˚C for 2 days. The plates were then analysed for the presence or absence of growth [30].

## 2.5 Activity of the extracts in combination with clinically used antifungals

The activity of free ESB in combination with fluconazole, nystatin or caspofungin were determined by the checkerboard assay as previously described [31]. Briefly, column 2–10 of a round-bottom 96-wellplate was filled with 50 μL RPMI medium containing L-glutamine and buffered with 165 mM morpholinepropanesulfonic acid (MOPS, pH 7.0). Drug A (fluconazole, nystatin or caspofungin) (100 μL) was added in column 1 and 50 μL serial 2-fold dilutions performed from column 1–9. In another plate, row B to H was filled with 50 μL of RPMI, and drug B (ESB, 100 μL) was added to wells in row A (column 1–10) and subsequently diluted 2-fold from row A to G. Afterwards, the different dilutions of drug B were transferred to the well-plate containing dilutions of drug A. Thus, row H (columns 1–9) and column 10 (rows A-H) contained drugs A and B alone, respectively. Fungal inoculum (100 μL in RMPI medium) was added from columns 1 to 11 (rows A–H) at a final concentration of $1 \times 10^3$ cells/mL, and the plates were incubated for 48 h at 37˚C. Columns 11 and 12 represent growth control (media + organism only) and negative control (media with no organism) controls respectively. The MIC values were determined by measuring the absorbance of the wells at 490 nm in a microtiter plate reader.

The result was analyzed by calculating the Fraction Inhibitory Concentration Indices (FICI) which was determined as: FICI = $FIC_A + FIC_B$, where $FIC_A = (MIC_{CA} / MIC_A)$ and $FIC_B = (MIC_{CB} / MIC_B)$. $MIC_A$ and $MIC_B$ are the Minimum Inhibitory Concentrations (MIC) of drugs A and B alone respectively, and $MIC_{CA}$ and $MIC_{CB}$ are the concentration of drugs A and B when used in combinations respectively.

The FIC Indices were interpreted as follows: Synergism (FICI ≤ 0.5), Indifference (> 0.5– 4.0) and Antagonism (> 4.0).

## 2.6 Activity of ESB against *Candida* biofilms

The free and formulated ESB activity against biofilms of the *Candida* strains under two different experimental conditions: biofilm formation inhibition and effect against mature biofilms was evaluated by the fungal biofilm formation and susceptibility testing in 96-well plates. The experiment was replicated thrice in both assays.

**2.6.1 Inhibition of biofilm formation assay.** Briefly, 50 μL RPMI 1640 was added to each well in a flat-bottom 96-well microplate together with 50 μL of the free extract in column 1. This was serially diluted till column 10 to obtain 1000–1.95 μg/mL concentrations. Caspofungin (16–0.31 μg/mL) was used a positive control. Thereafter, 50 μL of overnight cultured fungal suspension at a concentration of $2 \times 10^6$ cells/mL were added to wells of columns 1–11, and the plates were incubated at 37˚C for 24 h. Columns 11 and 12 represent growth control (media + organism only) and negative control (media with no organism) controls respectively. After incubation, the media in each well was carefully aspirated to not disrupt the biofilms and the plates were washed thrice with PBS (100 μL) to remove non-adherent and/or planktonic cells. Afterward, 100 μL of XTT/menadione was added to each well and the plates were incubated in the dark for 2 h at 37˚C. Then 80 μL of the resulting-coloured supernatant from each

well was transferred into new microplates and the absorbances of the plates were recorded at 490 nm using a microplate reader [32].

**2.6.2 Inhibition of mature biofilm assay.** Yeast inoculum (100 μL of a suspension of $1 \times 10^6$ cells/mL in RPMI 1640) of each *Candida* strain was added to wells of a flat-bottom 96-well microplate and incubated for a day at 37˚C to allow for biofilm formation. Medium from each well was removed carefully so as not to touch the biofilms formed and the wells were washed with 100 μL PBS (2×) to remove non-adherent and/or planktonic cells. Dilutions of the free and formulated extracts were prepared from 1500 to 5.85 μg/mL in another 96-well plate and transferred to the well plate containing the mature biofilms. Caspofungin (16– 0.31 μg/mL) was used a positive control. This was further incubated at 37˚C for a day. After-wards, the media in the wells were carefully removed and the plate was washed 2× with 100 μL PBS. XTT/menadione solution (100 μL) was added to each well and the plates were incubated at 37˚C for 2 h in the dark. Afterward, 80 μL of the resulting supernatant from each well was transferred into a new microplate and the absorbance measured at 490 nm [33].

## 2.7 Metabolite profiling

**2.7.1 Phytochemical characterization by UPLC-ESI-QTOF-MS/MS.** The phytochemical constituents in the ESB extract were characterised with a Dionex Ultimate 3000 RS Liquid Chromatography System. The separation was carried out on a Dionex Acclaim RSLC 120, C18 column (2.1 × 100 mm, 2.2 μm) with a binary gradient (A: water with 0.1% formic acid; B: ace-tonitrile with 0.1% formic acid) at 0.4 ml/min: isocratic at 5% B; 0.4 to 9.9 min: linear from 5% B to 100% B; 9.9 to 15.0 min: isocratic at 100% B; 15.0 to 15.1 min: linear from 100% B to 5% B; 15.1 to 20.0 min: isocratic at 5% B. The injection volume was 2 μL. The eluted phytoconsti-tuents were detected over the range 200–400 nm with a Dionex Ultimate DAD-3000 RS and a Bruker Daltonics micrOTOF-QII time-of-flight mass spectrometer equipped with an Apollo electrospray ionisation source in positive mode at 3 Hz over the mass range of *m/z* 50–1,500 using the following instrument settings: nebuliser gas nitrogen, 4 bar; dry gas nitrogen, 9 L/ min, 200˚C; capillary voltage −4,500 V; end plate offset −500 V; transfer time 100 μs, prepulse storage 6 μs, collision energy 8 eV. MS/MS scans were triggered by AutoMS2 settings within a range of *m/z* 200–1,500, using a collision energy of 40 eV and collision cell RF of 130 Vpp. Internal dataset calibration (HPC mode) was performed for each analysis using the mass spec-trum of a 10 mM solution of sodium formate in 50% isopropanol that was infused during LC re-equilibration using a divert valve equipped with a 20 μl sample loop. With the aid of the Reaxys® online database, the identities of the compounds were putatively assigned.

**2.7.2 UPLC fingerprinting for quality control.** The UPLC fingerprint profiles of the hydroethanolic extract of ESB was developed using the Acquity UPLC® (Waters, Milford, U. S.A.) system equipped with PDA eλ detector (200–400 nm); QDa detector (ESI, positive mode, single quadrupole, 100–600 Da); sample manager (inj.-vol.: 2 μL); column heater (40˚C); sta-tionary phase: Waters Acquity UPLC® HSS T3 (2.1 × 100 mm, 1.8 μm); Empower 3 Software; a binary solvent manager with a flow rate: 0.5 mL/min, and mobile phases: A: $H_2O$ + 0.1% for-mic acid, B: $CH_3CN$ + 0.1% formic acid in a gradient elution format. With a run time of 20 minutes, the elution system of ESB adopted the following scheme: 0–1 min, 2% B; 1–10 min, 2%-10% B; 10–20 min, 10%-15% B. The column was then washed with 100% B for the next 2 mins and then re-equilibrated to the initial conditions with additional 2 mins run with 2% B. The chromatogram was optimally detected at 280 nm and the compounds present were con-firmed from the corresponding mass spectral data to the peaks observed. For qualitative pur-poses, the Relative Retention Times (RRT) and Relative Peak Areas (RPA) of the prominent UV-absorbing compounds observed in the chromatograms (n = 18) were calculated, in

reference to vicenin-2 (m/z = 594.20; Rt = 12.56 mins), considered as an internal standard and forms part of the constituents of the extract.

The fingerprint was then validated following the ICH Q2(R1) guidelines [34]. Specificity, precision, and stability were the parameters considered. In evaluating specificity, the prominent phytochemicals in the chromatogram were identified through their mass spectral data generated. The precision of the fingerprint was evaluated by determining repeatability and intermediate precision. This was done by observing the RRTs and RPAs of seven of the prominent peaks from replicate analysis (n = 6) of the extracts on same and different days. The relative standard deviations were then calculated. The stability of the fingerprints was also determined over a 48-hour period at predetermined time intervals (0, 6, 12, 24 & 48 hours). The percentage change in the relative peak areas of the internal reference compound (vicenin-2) was then monitored.

## 3. Results

### 3.1 Activity of the free and formulated *E. senegalensis* extract against *C. albicans* and *C. glabrata* strains

The free *E. senegalensis* stem bark extract (ESB) and the extract incorporated in a microemulsion formulation exhibited strong to moderate antifungal activity (MIC = 3.91–31.25 µg/mL) against the strains of *C. albicans* and *C. glabrata* strain (Table 1). There were similar levels of bioactivity for the free and formulated extract against both *C. albicans* strains. Interestingly, both the free and formulated ESB demonstrated stronger activity against the *C. glabrata* strain.

As seen in Table 1, the MFC values (3.91–31.25 µg/mL) of the formulated extract were the same as its MIC values against strains of both *Candida* species. However, a much greater concentration of the free ESB was needed to exert a fungicidal effect.

### 3.2 Activity of free ESB against clinical isolates of *C. albicans* and *C. glabrata* exhibiting different degrees of resistance against conventional antifungals

The activity of the free extract against a series of *C. albicans* and *C. glabrata* clinical isolates recovered from pregnant women with VVC that are resistant to treatment with conventional antifungals are presented in S1 and S2 Tables in S1 File, respectively. The free extract exhibited very strong antifungal activity against all the *C. albicans* (8–64 µg/mL) and *C. glabrata* (4–16 µg/mL) isolates tested, irrespective of their resistance patterns and decreased susceptibility to the clinically used antifungals.

**Table 1. Inhibitory effect of free or formulated ESB against *C albicans* and *C. glabrata* strains.**

| Strain/Sample | C. albicans | | | | C. glabrata | |
| --- | --- | --- | --- | --- | --- | --- |
| | ATCC 10231 | | SC5314 | | ATCC 2001 | |
| | MIC | MFC | MIC | MFC | MIC | MFC |
| Free ESB | 15.63 | 125 | 31.25 | 250 | 7.81 | 62.5 |
| Formulated ESB | 15.63 | 15.63 | 31.25 | 31.25 | 3.91 | 3.91 |
| Voriconazole | 4.00 | - | 4.00 | - | 4.00 | - |
| Nystatin | 16.00 | - | 8.00 | - | 8.00 | - |

Values are in µg/mL. ESB: *E. senegalensis* 70% ethanol stem bark extract; MIC: Minimum Inhibitory Concentration; MFC: Minimum Fungicidal Concentration. Experiment was carried out in triplicates.

**Table 2. Effect of *E. senegalensis* stem bark extract on the antifungal activity of fluconazole, nystatin and caspofungin.**

| Specie | Strain | ESB in combination with | | | | | |
|---|---|---|---|---|---|---|---|
| | | Fluconazole | | Nystatin | | Caspofungin | |
| | | FICI | INT | FICI | INT | FICI | INT |
| *C. albicans* | **ATCC 10231** | 0.38 | S | 0.31 | S | 1.00 | I |
| | **SC5314** | 0.63 | I | 0.75 | I | 1.25 | I |
| *C. glabrata* | **ATCC 2001** | 0.24 | S | 0.50 | S | 1.00 | I |

ESB: *E. senegalensis* 70% ethanol stem bark extract.

## 3.3 Activity of ESB in combination with clinically used antifungal agents

The antifungal activity of fluconazole, nystatin and caspofungin against the *Candida* strains when combined with the free extract (ESB) was determined by the checkerboard assay. The combination of the extract with fluconazole and nystatin respectively resulted in synergism against *C. albicans* ATCC 10231 (FICI = 0.38; 0.31) and *C. glabrata* ATCC 2001 (FICI = 0.24; 0.50). However, that combination showed indifference in *C. albicans* SC5314 (Table 2). The combination of caspofungin with ESB produced only indifference interactions in both the *C. albicans* and *C. glabrata* strains.

## 3.4 Effect of the free and formulated *E. senegalensis* extract on biofilms of *C. albicans* and *C. glabrata*

The activity of *E. senegalensis* stem bark extract (ESB) and its microemulsion formulated extract against *C. albicans* and *C. glabrata* biofilms were evaluated under two different experimental conditions. We initially evaluated their tendencies to inhibit biofilm formation of the two *Candida* species. Thereafter, we evaluated their activities against the preformed biofilms. As seen in Table 3, the free and formulated ESB inhibited biofilm formation of the *Candida* species at concentrations (16–64 μg/mL) slightly higher than those required to inhibit the growth of planktonic cells. However, much greater concentrations are required to disrupt matured biofilms (128 ->512 μg/mL). The effect of the extract on the *Candida* biofilms was more pronounced in the *C. albicans* strains than in the *C. Glabrata* strain.

## 3.5 Phytochemical profiling

The phytochemical characterisation of the hydroethanolic extract of ESB by RP-HPLC–ESI–QTOF/MS$^2$ showed the predominance of phenolic and polyphenolic compounds. Some of the phenolic compounds detected included arylbenzofurans like Erythbidin D. The polyphenolics

**Table 3. Effect of free or formulated ESB on biofilms of *C albicans* and *C. glabrata* strains.**

| Specie | Strain | SMIC$_{50}$ (μg/mL) | | | | | |
|---|---|---|---|---|---|---|---|
| | | Free ESB | | Formulated ESB | | Caspofungin | |
| | | Inhibition | Preformed | Inhibition | Preformed | Inhibition | Preformed |
| *C. albicans* | **ATCC 10231** | 32 | 256 | 16 | 128 | 0.031 | 0.031 |
| | **SC5314** | 64 | 128 | 16 | 128 | 0.063 | 0.063 |
| *C. glabrata* | **ATCC 2001** | 64 | >512 | 32 | 512 | 0.063 | 0.063 |

SMIC: Sessile Minimum Inhibitory Concentration.

**Fig 1. Chemical structures of some compounds identified in the 70% ethanol stem bark extract of *E. senegalensis*.**

were mostly flavonoids and their prenylated derivates, including flavones, isoflavones, flavonones, isoflavonones, pterocarpans, among others. Two alkaloids, that is, Glucoerysodine and Erythbidin D were also detected. The compounds identified putatively are summarised in **Fig 1** and **S3 Table in S1 File**.

For quality control purposes, the UPLC fingerprint of the extract was developed. By considering the number of prominent peaks detected, peak symmetries and a balance between peaks and noise detection, the wavelength 280 nm was consequently adopted for the peak detection (**Fig 2**). The RRTs and RPAs of the 18 peaks selected were calculated and used as reference parameters in authenticating the extract (**Table 4**). The RRTs and RPAs were shown to be

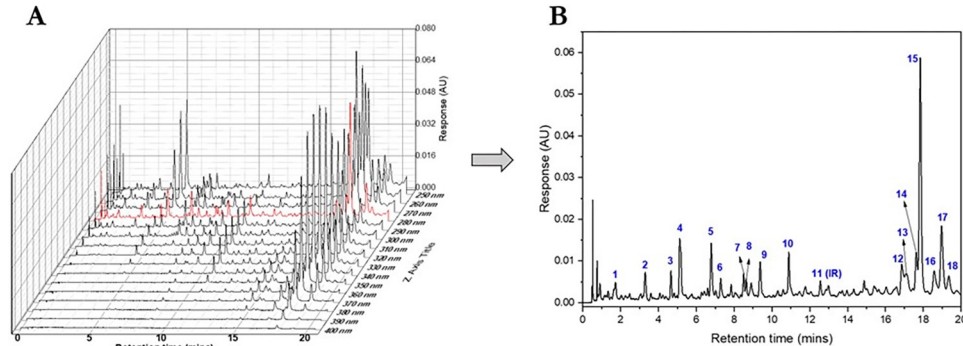

**Fig 2. Representative UPLC fingerprint profile for ESB extract.** [A]–Stack illustration of chromatograms of the extract at different wavelengths. The red highlighted chromatogram (that is, at 280 nm) was selected for the fingerprinting analysis. [B]–The fingerprint detected at 280 nm showing 20 prominent peaks used to identify the extract. The peak 11, which was identified to be vicenin-2 was selected as the internal reference (IR) peak.

precise from the determinations of repeatability and intermediate precision (**Table 5**). The RPA of the internal reference compound was also observed to be stable beyond 48 hours of leaving the test solution standing. The results therefore indicate that the ESB UPLC fingerprint developed and validated is suitable as a preliminary qualitative tool to authenticate the plant part used for medicinal purposes.

## 4. Discussion

The present study evaluated the antifungal activity of *E. senegalensis* ethanol stem bark extract (ESB) free or incorporated into a microemulsion formulation against *C. albicans* and *C. glabrata* strains and clinical isolates.

**Table 4. Relative retention times (RRT) and relative peak areas (RPA) for *E. senegalensis* plant extracts as determined for qualitative purposes.**

| Cpd ID | Ret time | Peak area | Relative retention time (RRT) | Relative peak area (RPA) |
|--------|----------|-----------|-------------------------------|--------------------------|
| 1 | 1.72 | 20477 | 0.14 | 1.1377 |
| 2 | 3.30 | 33557 | 0.26 | 1.8645 |
| 3 | 4.66 | 29475 | 0.37 | 1.6377 |
| 4 | 5.12 | 108019 | 0.41 | 6.0017 |
| 5 | 6.79 | 71948 | 0.54 | 3.9976 |
| 6 | 7.29 | 30005 | 0.58 | 1.6671 |
| 7 | 8.50 | 27742 | 0.68 | 1.5414 |
| 8 | 8.64 | 19861 | 0.69 | 1.1035 |
| 9 | 9.38 | 57080 | 0.75 | 3.1715 |
| 10 | 10.89 | 41804 | 0.87 | 2.3227 |
| 11* | 12.56 | 17998 | 1.00 | 1.0000 |
| 12 | 16.86 | 71255 | 1.34 | 3.9591 |
| 13 | 17.09 | 51630 | 1.36 | 2.8687 |
| 14 | 17.63 | 64695 | 1.40 | 3.5946 |
| 15 | 17.84 | 424560 | 1.42 | 23.5893 |
| 16 | 18.58 | 65207 | 1.48 | 3.6230 |
| 17 | 18.97 | 164917 | 1.51 | 9.1631 |
| 18 | 19.35 | 47261 | 1.54 | 2.6259 |

*Compound 11 identified as Vicenin-2 was used as an internal reference marker for ESB extract.

**Table 5. Validation parameters for UPLC Fingerprint profiling of ESB sample.**

| Peak No | Precision | | | | Stability (over 48 hours) ΔRPA (%) |
|---|---|---|---|---|---|
| | Repeatability (same day) (n = 6) | | Intermediate precision (different days) (n = 9) | | |
| | Mean RRT, RSD (%) | Mean RPA, RSD (%) | Mean RRT, RSD (%) | Mean RPA, RSD (%) | |
| 4 | 0.41 ± 0.004, 1.00 | 6.062 ± 0.077, 1.26 | 0.41 ± 0.003, 0.82 | 6.000 ± 0.129, 2.15 | Peak 11 –Vicenin-2: 1.47 |
| 5 | 0.50 ± 0.000, 0.00 | 4.012 ± 0.055, 1.36 | 0.54 ± 0.000, 0.00 | 3.991 ± 0.058, 1.44 | |
| 10 | 0.87 ± 0.005, 0.63 | 2.339 ± 0.031, 1.33 | 0.86 ± 0.005, 0.61 | 2.323 ± 0.054, 2.34 | |
| *11 | 1.00 ± 0.000, 0.00 | 1.000 ± 0.000, 0.00 | 1.00 ± 0.000, 0.00 | 1.000 ± 0.000, 0.00 | |
| 15 | 1.42 ± 0.000, 0.00 | 23.68 ± 0.292, 1.24 | 1.42 ± 0.003, 0.23 | 23.35 ± 0646, 2.77 | |
| Acceptance criteria | RSD < 2% | RSD < 5% | RSD < 2% | RSD < 5% | % ΔRPA < 5% |

*Internal reference marker compound.

The free and formulated ESB exerted strong *in vitro* antifungal activity against the collection strains of *C. albicans* and *C. glabrata* with MIC ranging from 3.91 to 31.25 µg/mL. The free and formulated extracts exhibited similar levels of activity against both *C. albicans* strains. Interestingly, they demonstrated a higher antifungal activity against *C. glabrata* (**Table 1**). The minimum fungicidal concentrations (MFCs) of the free and formulated extracts against the same collection strains were also determined. Whereas higher concentrations of the free extract were needed to exert fungicidal activity against both *Candida* species, the MFCs of the formulated extract were the same as their corresponding MIC values suggesting a more potent antifungal activity for the extract when loaded into the microemulsion formulation than free.

The emergence of resistance to currently used antifungals has become a major clinical challenge limiting the effective treatment of *Candida* infections. In *C. albicans*, prolonged use and overexposure typically leads to the development of azole resistance, whereas *C. glabrata* is intrinsically resistant to azoles, with resistance to polyenes and echinocandins emerging in recent years [35,36]. This further underscore the need for the discovery and development of new antifungal drugs, particularly those active against resistant species and strains. Thus, the free extract was evaluated for antifungal activity against a series of *C. albicans* and *C. glabrata* clinical isolates recovered from pregnant women with VVC that are resistant to azoles or polyenes (**S1 and S2 Tables in S1 File**). The free ESB demonstrated strong activity against all the *C. albicans* and *C. glabrata* isolates tested, which was completely unaffected by their resistance patterns to conventional antifungals. Therefore, the stem bark extract of *E. senegalensis* may represent a promising alternative to the treatment of Candida infections caused by *C. albicans* and *C. glabrata* including those that may fail conventional treatment.

One of the therapeutic strategies to overcome antifungal resistance is the use of combination therapy. This strategy has been routinely utilised in the treatment of diverse infectious diseases including tuberculosis, malaria and HIV/AIDS, while its potential in antifungal treatment is only beginning to be explored [37,38]. The advantage being specific drug combinations can overcome existing resistance and/or delay or prevent further resistance from occurring. For instance, by combining multiple antifungals with different mechanisms of action, the number of intracellular processes targeted increases, which makes the development of resistance more difficult as the *Candida* specie or strain must acquire mutations in multiple genes [39]. Furthermore, antifungal combination therapy can increase the fungicidal efficiency of normally fungistatic drugs reducing *Candida* specie or strain population sizes and subsequently the probability of resistance emerging [40]. Thus, the antifungal activity of fluconazole, nystatin and caspofungin against the *Candida* strains in combination with the extract (ESB) was determined by the checkerboard assay. After calculating their respective FICI values, the

combination of ESB with caspofungin showed only indifference interactions in all the *Candida* strains, however, its combinations with fluconazole or nystatin resulted in synergistic antifungal effect against *C. albicans* ATCC 10231 and *C. glabrata* ATCC 2001. The results indicate the extract, and its compounds could be a source of newer antifungal agents with novel mechanisms of actions that could be used alone or in combination with existing antifungals for the development of antifungal combination therapy for treating drug-resistant fungal infections.

The pathogenicity of *Candida* species is attributed to certain virulence factors, one of which is their ability to form biofilms [41]. *Candida* biofilms exhibit phenotypically distinct traits from those of their planktonic counterparts. Particularly, *Candida* cells embedded within biofilms demonstrate less sensitivity to conventional antifungals and have survival advantage over planktonic cells [42]. It was therefore not surprising that higher concentrations ($SMIC_{50}$ = 128 −>512 μg/mL) of the extract whether free or loaded in the microemulsion formulation were required to inhibit the mature biofilms compared to the planktonic cells (MIC = 3.91–31.25 μg/mL). On the other hand, the free and formulated extract inhibited biofilm formation at concentrations ($SMIC_{50}$ = 16–64 μg/mL) slightly higher than those inhibiting the planktonic cells. Generally, biofilms of *C. glabrata* strain were more resistant to the extract than the biofilms of the *C. albicans* strains and the formulated extract exhibited more potent activities than the free extract.

The observed antifungal activity of ESB against *C. albicans* and *C. glabrata* could partly be explained by the presence of the phytochemicals in the hydroethanolic extract. Although several compounds were not identified in the study and considered novel, the extract contained a number of phenolic and polyphenolic compounds as well as alkaloids, several of which have been established to possess antimicrobial activities against different microorganisms [43]. The presence of polyphenolic compounds in the 70% ethanol extract particularly isoflavonoids were also detected in methanol and ethyl acetate extracts of the stem bark of the plant [17]. For the flavonoids particularly, it has been observed that the presence of the prenylated or 2,2-dimethylpyrano ring substituents positively impact on their biological activities [44]. With respect to antifungal effects, Neobavaisoflavone has been shown to be active against *Cryptococcus neoformans* and *Aspergillus fumigatus* [45]. Alpinumisoflavone, Cristacarpin and 6,8-diprenylgenistein on the other hand showed antifungal activities against *C. mycoderma* [46,47]. The apparent presence of these compounds in addition to others could be thought to contribute to the observed antifungal activity against *C. albicans* and *C. glabrata* in the current study.

Since the plant possess numerous ethnopharmacological uses and in the current study, it has been shown to be active against resistant *Candida* species, its quality control had become even more important. Also, since many of the constituents identified in ESB occur in other similar species, adopting single marker assessments for quality control may be inadequate. Hence, in this study, we adopted the fingerprint profiling approach, where different species of *Erythrina* may likely possess different fingerprint patterns depending on the differences in their phytocompositions. For such reason, the authentication of ESB could be ensured by its characteristic discriminating fingerprint pattern from the other species and can thus, be routinely adopted. We have hence reported the RRTs of peaks in ESB and their corresponding RPAs (**Table 4**) that can serve as qualitative information in the quality control of the plant. The validation outcomes confirm the reproducibility of the approach and hence propose its adoption.

## 5. Conclusion

In summary, the present study has demonstrated the antifungal activity of the stem bark extract of *E. senegalensis*, when used both as a free extract or incorporated into a

microemulsion formulation. Compared to its activity against *C. albicans*, the extract exhibited a stronger activity against *C. glabrata*. Furthermore, the extract was active against several clinical isolates of *C. albicans* and *C. glabrata* that showed resistance against clinically used antifungal agents and could be used in combination with conventional antifungals to enhance the efficacy of treatment even against drug-resistant fungal infections. The extract also considerably inhibited biofilm formation as well as disrupted mature biofilms of both *Candida* species. These observed activities could partly be explained by the presence of compounds like Neobavaisoflavone, Alpinumisoflavone, Cristacarpin and 6,8-diprenylgenistein among other prenylated flavonoids and alkaloids in the extract. It is however recommended that *in vivo* studies be carried to confirm the observed anti-candida effects. A UPLC fingerprint has been developed and validated and proposed to be for the routine assessment of the plant due to its numerous ethnopharmacological importance.

## Supporting information

**S1 Fig. HPLC chromatogram of 70% ethanol stem bark extract of E. senegalensis from the UPLC-ESI-QTOF-MS2 analysis.**
(TIF)

**S1 File.**
(DOCX)

## Acknowledgments

We are grateful to Dr. Shirley Ocloo and the Ghana Physicians and Surgeons Association, North America (GPSA-NA, N&S Carolina Chapter) for furnishing the Organic and Analytical laboratory, SBBS, UHAS where the work was done. The team is also grateful to Seeding Labs, Boston, US for the equipment used in the study and to the Microbiology laboratory of the Ho teaching Hospital, Ho for providing the *C. albicans and C. glabrata* clinical isolates.

## Author Contributions

**Conceptualization:** Benjamin Kingsley Harley, Mike Okweesi Aggrey, Theophilus Christian Fleischer.

**Data curation:** Benjamin Kingsley Harley, Anthony Martin Quagraine, David Neglo, Emmanuel Orman.

**Formal analysis:** Benjamin Kingsley Harley, Anthony Martin Quagraine, David Neglo, Emmanuel Orman, Cedric Dzidzor Amengor, Jonathan Jato, Yussif Saaka, Theophilus Christian Fleischer.

**Investigation:** Benjamin Kingsley Harley, Anthony Martin Quagraine, Mike Okweesi Aggrey, Emmanuel Orman, Cedric Dzidzor Amengor, Jonathan Jato, Yussif Saaka.

**Methodology:** Benjamin Kingsley Harley, Anthony Martin Quagraine, David Neglo, Jonathan Jato, Yussif Saaka, Theophilus Christian Fleischer.

**Project administration:** Benjamin Kingsley Harley, Mike Okweesi Aggrey.

**Supervision:** Benjamin Kingsley Harley, Mike Okweesi Aggrey, Theophilus Christian Fleischer.

**Validation:** Benjamin Kingsley Harley, Mike Okweesi Aggrey, Emmanuel Orman.

**Writing – original draft:** Benjamin Kingsley Harley, Emmanuel Orman, Nana Ama Mireku-Gyimah.

**Writing – review & editing:** Benjamin Kingsley Harley, Nana Ama Mireku-Gyimah, Cedric Dzidzor Amengor, Theophilus Christian Fleischer.

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
