## [Decision Letter · Decision Letter 0]

26 Jul 2022

PONE-D-22-14963UPLC-ESI-QTOF-MS2 metabolite profiling, antifungal, biofilm formation prevention and disruption of mature biofilm activities of Erythrina senegalensis stem bark extract against Candida albicans and Candida glabrataPLOS ONE

Dear Dr. Harley,

Thank you for submitting your manuscript to PLOS ONE. After careful consideration, we feel that it has merit but does not fully meet PLOS ONE’s publication criteria as it currently stands. Therefore, we invite you to submit a revised version of the manuscript that addresses the points raised during the review process.

We look forward to receiving your revised manuscript.

Kind regards,

Umakanta Sarker

Academic Editor

PLOS ONE

Journal Requirements:

   "We are grateful to Dr. Shirley Ocloo and the Ghana Physicians and Surgeons Association, North America (GPSA-NA, N&S Carolina Chapter) who provided funds for furnishing the Organic and Analytical laboratory, SBBS, UHAS where the work was done. The team is also grateful to Seeding Labs, Boston, US for the provision of some equipment used in the study and to the Microbiology laboratory of the Ho teaching Hospital, Ho for providing the C. albicans and C. glabrata clinical isolates."

Reviewers' comments:

Reviewer's Responses to Questions

**Comments to the Author**

1. Is the manuscript technically sound, and do the data support the conclusions?

Reviewer #1: Yes

Reviewer #2: Partly

Reviewer #3: Yes

2. Has the statistical analysis been performed appropriately and rigorously? 

Reviewer #1: N/A

Reviewer #2: No

Reviewer #3: Yes

3. Have the authors made all data underlying the findings in their manuscript fully available?

Reviewer #1: Yes

Reviewer #2: No

Reviewer #3: Yes

4. Is the manuscript presented in an intelligible fashion and written in standard English?

Reviewer #1: Yes

Reviewer #2: No

Reviewer #3: Yes

5. Review Comments to the Author

Reviewer #1: The following comments needs to be addressed before acceptance:

• Recommended to add the reference for the zone inhibition line 145-150

• Age of the fungal suspension used for “2.6.1 Inhibition of biofilm formation assay line 199”

• Justify why the authors used solvent 70% ethanol, why not different concentrations and other solvents.

• There are many references available the phenolic compounds such as flavonoids and alkaloids from plant sources has the ability to inhibit the fungal pathogens especially yeasts. The authors need to suggest the novelty of this research by highlighting a potential antifungal compound (S) based on UPLC fingerprints or any new compound(s) which are not reported before.

• The write-up flow of the article is scientifically well described and easy to follow. The methods used by the authors are up to date.

• Further the authors are recommended to read and cite the article to substantiate this research.

Al Aboody, M. S., & Mickymaray, S. (2020). Anti-fungal efficacy and mechanisms of flavonoids. Antibiotics, 9, 45. https://doi.org/10.3390/antibiotics9020045

Mickymaray, S.; Alturaiki, W. (2018). Antifungal efficacy of marine macroalgae against fungal isolates from bronchial asthmatic cases. Molecules, 23, 3032. https://doi.org/10.3390/molecules23113032.

https://doi.org/10.1016/j.apjtb.2015.12.005

Best regards

Reviewer #2: The manuscript by Harley et al attempted to evaluate the anti-fungal activities and phytochemical profile of Erythrina senegalensis stem bark extracts. Despite its objectives, the manuscript is poorly organized, lacks coherence, and hence, needs extensive revision regarding its grammar and composition. Besides, there are several scientific issues that the authors should give due attention to. Some of the major issues are stated below.

•The title is too long and can be rewritten. Likewise, the abstract is very broad and hence, should be recomposed to be clear and precise.

•The plant, Erythrina senegalensis, is one of the most widely studied species. Several previous studies investigated the phytochemical compositions and pharmacological properties of the different tissues of the plant. For instance, recently, phytochemicals from the stem bark of the plant were investigated using LC-MS (Molecules 2022, 27, 2583). Therefore, the authors should spend some time searching literature and acknowledge those previous findings (both in the introduction and discussion parts). Most importantly, they should indicate the research gap they want to address in their study.

•Section 2.3: The experiment/procedure stated here looks incomplete. The authors should clarify how they treated the final extract (after sonication) to obtain the target microemulsion extract (what makes it "microemulsion"?).

•Section 2.4.3: The microbroth dilution protocol for the anti-fungal assay is poorly described. Fungal growth, cell collection, dilution procedure, amount of extracts applied, etc should be properly explained. Overall, the anti-fungal assays (sections 2.4, 2.5, and 2.6) needs revision and should be clearly presented. I recommend the authors to review some similar literature.

•Sections 3.1, 3.2, and 3.3 should reflect/contain some numerical values.

•The authors conducted the profiling of phytochemicals (section 2.7.1) and the validation (section 2.7.2) in separate instruments using different analysis conditions (methods). This is not clear and unacceptable. I do not think method development and validation are the motives of this paper, and hence, I recommend the removal of the validation part. Or else, the target metabolite should be properly quantified (using standards) and all the validation parameters should be fully considered.

•What are the identities of the compounds given in Table 4 (and as peak numbers in Figure 2B)? Are they similar to those given in Supplementary Table S3? Why do similar molecules show different retention times (RT)? For instance, Vicenin-2 has RT of 4.86 in Table S3, and 12.56 in Table 4.

•There are several unidentified major peaks (Supplementary Figure S1). It is not clear why the authors failed to do so.

•I don’t see the value of adding figure 2A. The manuscript is short of illustrations and figures. I recommend the authors incorporate some from their experimental findings.

•The m/z for fragment molecules from the MS/MS is provided only for a few compounds (S3 Table). The observed major MS/MS fragments for all the compounds should be provided. I recommend this table (showing the list of all the compounds including the currently unidentified peaks) be part of the main body of the manuscript.

•In the reference section, the name of authors and editors, pages, publishers, etc should be provided for cited books.

Reviewer #3: The authors investigated “UPLC–ESI–QTOF-MS2 Metabolite profiling, antifungal, biofilm formation prevention and disruption of mature biofilm activities of Erythrina senegalensis stem bark extract against Candida albicans and Candida glabrata”. The manuscript falls into the scope of the journal. The manuscript is well-written and contains valuable information. The fluidity is steady, the content is meaningful and understandable, and the structure is considerably organized. The authors present adequate information in the methodology to allow the reproduction of the estimated methods properly. I have just a few questions/comments.

1) Delete UPLC–ESI–QTOF-MS2 from the title.

2) Line # 13 : Write full name of E. senegalensis and also include English name.

3) Line # 14: Write full name: C. albicans and C. glabrata

4) Line # 15-19: Simplify the sentences.

5) Line # 22: Write full name MIC

6) Line # 23: Write full name MFC

7) Line # 51: Briefly introduce the pathogenicity factors of Candida and then describe biofilm as a pathogenicity factor.

8) Line # 89-90: An internet search revealed a scarcity of information on the antifungal activity of the plant and specifically on its anti-Candida activity.

9) Line # 153: Define free extract

10) Line # 174: Include the name of the conventional antifungal.

11) Line # 182: represent growth and media controls. Clarify growth.

12) Line # 186: 1 × 103 cells/mL or spores/mL???

13) Line # 192-194: Not clear. What kind of design you used here.

14) Line # 201: Biofilm not only forms in the surfaces but also found in the air-liquid interface. How about Candida spp.???

15) Line # 207: How do you compare biofilm inhibition? Where the control set?

6. PLOS authors have the option to publish the peer review history of their article (what does this mean?). If published, this will include your full peer review and any attached files.

Reviewer #1: **Yes: **Dr. Suresh Mickymaray Ph.D.,

Reviewer #2: No

Reviewer #3: **Yes: **Dr. Md. Manjurul Haque, Professor, Department of Environmental Science, Faculty of Agriculture, Bangabandhu Sheikh Mujibur Rahman Agricultural University, Gazipur 1706, Bangladesh

---

## [Author Response · Author response to Decision Letter 0]

3 Oct 2022

Academic Editor 

1. Please ensure that your manuscript meets PLOS ONE's style requirements, including those for file naming. ThePLOS ONE style templates can be found at https://journals.plos.org/plosone/s/file?id=wjVg/PLOSOne_formatting_sample_main_body.pdf

And https://journals.plos.org/plosone/s/file?id=ba62/PLOSOne_formatting_sample_title_authors_affiliations.pdf

We have ensured that the manuscript meets PLOS ONE’s style requirement for those for file naming

"We are grateful to Dr. Shirley Ocloo and the Ghana Physicians and Surgeons Association, North America (GPSA-NA, N&S Carolina Chapter) who provided funds for furnishing the Organic and Analytical laboratory, SBBS, UHAS where the work was done. The team is also grateful to Seeding Labs, Boston, US for the provision of some equipment used in the study and to the Microbiology laboratory of the Ho teaching Hospital, Ho for providing the C. albicans and C. glabrata clinical isolates."

Any funding related text has been removed from the text. The authors did not receive any funding for the study.

We do not want to update our funding statement.

Reviewer 1

1. Recommended to add the reference for the zone inhibition line 145-150

Reference for the zone inhibitions has been added

Reference number [25]

Line 149

2. Age of the fungal suspension used for “2.6.1 Inhibition of biofilm formation assay line 199”

The fungal suspension employed was prepared overnight. We have included in that in the manuscript.

Line 199

3. Justify why the authors used solvent 70% ethanol, why not different concentrations and other solvents.

70% ethanol was used in the study to mimic traditional methods for preparation of the plant material as medicine. We have included this explanation into the manuscript.

Lines 109-110

4 There are many references available the phenolic compounds such as flavonoids and alkaloids from plant sources has the ability to inhibit the fungal pathogens especially yeasts. The authors need to suggest the novelty of this research by highlighting a potential antifungal compound (s) based on UPLC fingerprints or any new compound(s)which are not reported before.

Indeed, a number of publications are available that highlight the phenolic compounds such as flavonoids and alkaloids. In our discussion, Section 4.0 lines 60-66, we mentioned a few of such compounds and referenced the literature. 

We have also included suggested reference, reference [42], section 4.0 Discussion line 59.

There were a number of unidentified peaks based on their MS data which we believe are novel. We indicated that in the manuscript and are even in the process of isolating and testing them as further work. 

On the issue of highlighting a potential antifungal compound based on UPLC fingerprint, the authors are of the view that the identified and non-identified compounds working additively or synergistically could produce the reported activity of the extract in the study. We also highlighted some of the identified compounds which have been shown to possess antifungal activities in other studies. 

Another point of novelty is that this is the first report of the antifungal activity of the plant against resistant Candida strains which is a phenomenon currently on the rise. 

Reviewer 2

The manuscript by Harley et al attempted to evaluate the anti-fungal activities and phytochemical profile of Erythrina senegalensis stem bark extracts. Despite its objectives, the manuscript is poorly organized, lacks coherence, and hence, needs extensive revision regarding its grammar and composition. Besides, there are several scientific issues that the authors should give due attention to. 

We strongly disagree with the assertion that the manuscript is poorly organized, lacks coherence, and hence, needs extensive revision regarding its grammar and composition. Based on comments of the academic editor and the other reviewers, we cannot accept such biased review and comment. The manuscript was proof-read and flows logically.

We also intend to address all scientific issues raised to enhance and improve the quality of the manuscript

1. The title is too long and can be rewritten. 

We have shortened the title by deleting UPLC–ESI–QTOF-MS2. 

2. Abstract is very broad and hence, should be recomposed to be clear and precise.

The abstract has been revised to make it clear and precise.

•The plant, Erythrina senegalensis, is one of the most widely studied species. Several previous studies investigated the phytochemical compositions and pharmacological properties of the different tissues of the plant. For instance, recently, phytochemicals from the stem bark of the plant were investigated using LC-MS (Molecules 2022, 27, 2583).

Therefore, the authors should spend some time searching literature and acknowledge those previous findings (both in the introduction and discussion parts). Most importantly, they should indicate the research gap they want to address in their study.

Indeed, Erythrina senegalensis is a widely investigated plant. The authors in the introduction cited and acknowledged previous phytochemical and pharmacological investigations, lines 79- 86.

It is rather unfortunate we were not able to reference the publication indicated above which was recently published a month before the manuscript was submitted. This might have resulted in its absence in the manuscript. We have duly cited and referenced it in the introduction and discussion.

On the research gap being addressed, the authors clearly stated that despite the numerous pharmacological Investigations done on the plant, its antifungal activity has yet to be evaluated, line 88 - 89. That is the research gap addressed. In Ghana, the stem bark of the plant is used in the treatment of infections including candidiasis in women and that prompted the study.

3. Section 2.4.3: The microbroth dilution protocol for the anti-fungal assay is poorly described. Fungal growth, cell collection, dilution procedure, amount of extracts applied, etc should be properly explained. Overall, the anti-fungal assays (sections 2.4, 2.5, and 2.6) needs revision and should be clearly presented. I recommend the authors to review some similar literature.

We have included the full protocol employed in evaluating the antifungal activity of the extract. Section 2.4.3 line 159 -170.

The assays presented in sections 2.5 and 2.6 were adapted from established protocols which have been used and cited severally in different publications. Concentrations of extracts employed are stated, amount of fungal inoculum employed are clearly stated. 

4. Sections 3.1, 3.2, and 3.3 should reflect/contain some numerical values.

Numerical values have been added to the sections stated 

Section 3.2 line 299 – 300.

Section 3.3 line 307 – 308. 

Section 3.1 already had numerical values present. Line 280

5. The authors conducted the profiling of phytochemicals (section 2.7.1) and the validation (section 2.7.2) in separate instruments using different analysis conditions (methods). This is not clear and unacceptable. I do not think method development and validation are the motives of this paper, and hence, I recommend the removal of the validation part. Or else, the target metabolite should be properly quantified (using standards) and all the validation parameters should be fully considered.

The LC-MS/MS characterisation and the UPLC fingerprinting serve different purposes: The characterisation seeks to comprehensively explore the phytocomposition of the plants, to understand the phytochemicals and appreciate the role they play in the biological effects investigated. The UPLC fingerprinting on the other hand seeks to develop conditions that could be replicated on the routine basis to qualitatively assess the plant. So, the UPLC fingerprinting is more a quality control tool for subsequent adoption. In developing the two methods, depending on the objective, the conditions may be similar or not. What is important is that the method is properly validated and can be replicated by anyone else. Before using the plant in any test (although it is botanically authenticated), it would be important to confirm its identity with another layer of check, and that is what the fingerprinting analysis provides. This procedure has been widely adopted for medicinal plants in Chinese Herbal Medicine and think its adoption for this plant would be beneficial. As such, authors hold the view that the quality control part of this work is as important as the biological testing part. Together, the two parts of this provide a comprehensive report that could be useful.

Fingerprinting analysis is more of a qualitative analysis than quantitative. The fingerprint profile seeks to be used to predict the presence (and not necessarily the quantity) of the compounds which could be identified by their relative retention times. In validating fingerprints for identification, parameters such as accuracy, linearity, LOD, LOQ are not required because it is not meant for quantitative purpose.

6. What are the identities of the compounds given in Table 4 (and as peak numbers in Figure 2B)? Are they similar to those given in Supplementary Table S3? Why do similar molecules show different retention times (RT)? For instance, Vicenin-2 has RT of 4.86 in Table S3, and 12.56 in Table 4.

The compounds given in Table 4 are the ones observed in the UPLC fingerprint developed for the purpose of routine identification for the plant. They are not similar (in numbering) to the ones provided in Table S3 (which were observed from the LC-MS analysis) because different chromatographic conditions were used for them. It is for that reason Vicenin-2 has different retention times in both conditions. The expectation is that all the compounds seen in Table 4 (subset) should also be present in Table S3 (universal set) but not in the same order because the conditions were different.

7. There are several unidentified major peaks (Supplementary Figure S1). It is not clear why the authors failed to do so.

Compounds were putatively annotated by comparing their mass spectral data with credible spectral libraries. There was no isolation of compounds for comprehensive characterization studies. In instances where spectral data for a peak matches with several compounds (for e.g. isomers), it was not practical to assign an identity to that peak in the absence of other data like NMR, IR, X-ray crystallography, etc. Hence the occurrence of several unassigned peaks. There are several instances of such in scientific literature.

8. I don’t see the value of adding figure 2A. 

Figure 2A shows the UPLC fingerprints developed at different wavelengths. In developing an hplc/uplc fingerprint, it is preferable to adopt conditions that lead to the clear detection of as many compounds as possible (the idea of achieving something close to the ideal). At different wavelengths, compounds may have different absorptivities and this could lead to poor signals at some wavelengths as compared to others. The figure, therefore, shows the signals for the identified separated compounds and the selected one (optimized wavelength) in red shows which wavelength produced a good balance of signal detection for all the compounds.

9. The manuscript is short of illustrations and figures. I recommend the authors incorporate some from their experimental findings.

10. The m/z for fragment molecules from the MS/MS is provided only for a few compounds (S3 Table). The observed major MS/MS fragments for all the compounds should be provided. I recommend this table (showing the list of all the compounds including the currently unidentified peaks) be part of the main body of the manuscript.

Ideally, all the compounds putatively identified should present with their MS/MS fragments, but not all the compounds had their MS/MS showing up in the experimental run. 

11. In the reference section, the name of authors and editors, pages, publishers, etc should be provided for cited books.

The publishers and pages of the cited books have been provided.

Reviewer 3

1) Delete UPLC–ESI–QTOF-MS2 from the title.

UPLC–ESI–QTOF-MS2 has been deleted from the title

2) Line # 13 : Write full name of E. senegalensis and also include English name.

The name of the plant has been written in full. 

3) Line # 14: Write full name: C. albicans and C. glabrata

The names of the Candida strains have been written in full

4) Line # 15-19: Simplify the sentences.

The sentence has been simplified.

5) Line # 22: Write full name MIC

MIC has been written in full

6) Line # 23: Write full name MFC

MFC has been written in full

7) Line # 51: Briefly introduce the pathogenicity factors of Candida and then describe biofilm as a pathogenicity factor.

The pathogenicity factors of Candida have been included. Line 51-56.

8) Line # 89-90: An internet search revealed a scarcity of information on the antifungal activity of the plant andspecifically on its anti-Candida activity.

The statement has been deleted.

9) Line # 153: Define free extract

The statement has been re-written to bring definition to the free

10) Line # 174: Include the name of the conventional antifungal.

The names of the conventional antifungals have been included

11) Line # 182: represent growth and media controls. Clarify growth.

Clarification has been given to the nature of the controls

12) Line # 186: 1 × 103 cells/mL or spores/mL???

We used fungal cells for the experiment. 

13) Line # 192-194: Not clear. What kind of design you used here.

Line # 192-194 is a preamble to the sections 2.6.1 and 2.6.2. The authors were indicating that two (2) approaches were used to evaluate the antibiofilm activity of the plant extract; one being biofilm formation inhibition and the other inhibition of mature biofilms.

14) Line # 201: Biofilm not only forms in the surfaces but also found in the air-liquid interface. How about Candida spp.???

Indeed, Candida spp biofilms also form on the air-liquid interface. However, per the nature of the protocol, focus was on the biofilms that were formed on the inner walls of the wells as at a point in time, the media and all planktonic and biofilms in the media would be aspirated. 

15) Line # 207: How do you compare biofilm inhibition? Where the control set?

Caspofungin was used as positive controls in the antibiofilm assays, and we have included that in the protocols. Line 214 and 230. 

Columns 11 and 12 of the wells in all the assays were growth control (media + organism only) and negative control (media with no organism) controls respectively.

---

## [Decision Letter · Decision Letter 1]

27 Oct 2022

PONE-D-22-14963R1UPLC-ESI-QTOF-MS2 metabolite profiling, antifungal, biofilm formation prevention and disruption of mature biofilm activities of Erythrina senegalensis stem bark extract against Candida albicans and Candida glabrataPLOS ONE

Dear Dr. Harley,

Thank you for submitting your manuscript to PLOS ONE. After careful consideration, we feel that it has merit but does not fully meet PLOS ONE’s publication criteria as it currently stands. Therefore, we invite you to submit a revised version of the manuscript that addresses the points raised during the review process. Please submit your revised manuscript by Dec 11 2022 11:59PM. If you will need more time than this to complete your revisions, please reply to this message or contact the journal office at plosone@plos.org. Please include the following items when submitting your revised manuscript:A rebuttal letter that responds to each point raised by the academic editor and reviewer(s). You should upload this letter as a separate file labeled 'Response to Reviewers'.A marked-up copy of your manuscript that highlights changes made to the original version. You should upload this as a separate file labeled 'Revised Manuscript with Track Changes'.An unmarked version of your revised paper without tracked changes. You should upload this as a separate file labeled 'Manuscript'.If applicable, we recommend that you deposit your laboratory protocols in protocols.io to enhance the reproducibility of your results. Protocols.io assigns your protocol its own identifier (DOI) so that it can be cited independently in the future. For instructions see: https://journals.plos.org/plosone/s/submission-guidelines#loc-laboratory-protocols. Additionally, PLOS ONE offers an option for publishing peer-reviewed Lab Protocol articles, which describe protocols hosted on protocols.io. Read more information on sharing protocols at https://plos.org/protocols?utm_medium=editorial-email&utm_source=authorletters&utm_campaign=protocols.

We look forward to receiving your revised manuscript.

Kind regards,

Umakanta Sarker

Academic Editor

PLOS ONE

Journal Requirements:

Additional Editor Comments:

According to journal style, the highlight must be removed from the manuscript. follow the journal style for text, tables and figures.

Reviewers' comments:

Reviewer's Responses to Questions

**Comments to the Author**

1. If the authors have adequately addressed your comments raised in a previous round of review and you feel that this manuscript is now acceptable for publication, you may indicate that here to bypass the “Comments to the Author” section, enter your conflict of interest statement in the “Confidential to Editor” section, and submit your "Accept" recommendation.

Reviewer #3: All comments have been addressed

2. Is the manuscript technically sound, and do the data support the conclusions?

Reviewer #3: Yes

3. Has the statistical analysis been performed appropriately and rigorously? 

Reviewer #3: Yes

4. Have the authors made all data underlying the findings in their manuscript fully available?

Reviewer #3: Yes

5. Is the manuscript presented in an intelligible fashion and written in standard English?

Reviewer #3: Yes

6. Review Comments to the Author

Reviewer #3: 1. The study presents the results of original research.

2. Results reported have not been published elsewhere.

3. Experiments, statistics, and other analyses are performed to a high technical standard and are described in sufficient detail.

4. Conclusions are presented in an appropriate fashion and are supported by the data.

5. The article is presented in an intelligible fashion and is written in standard English.

6. The research meets all applicable standards for the ethics of experimentation and research integrity.

7. The article adheres to appropriate reporting guidelines and community standards for data availability.

7. PLOS authors have the option to publish the peer review history of their article (what does this mean?). If published, this will include your full peer review and any attached files.

Reviewer #3: No

---

## [Author Response · Author response to Decision Letter 1]

9 Nov 2022

All responses have been included in the rebuttal letter

---

## [Editor Report · Decision Letter 2]

10 Nov 2022

Metabolite profiling, antifungal, biofilm formation prevention and disruption of mature biofilm activities of Erythrina senegalensis stem bark extract against Candida albicans and Candida glabrata

PONE-D-22-14963R2

Dear Dr. Harley,

We’re pleased to inform you that your manuscript has been judged scientifically suitable for publication and will be formally accepted for publication once it meets all outstanding technical requirements.

Kind regards,

Umakanta Sarker

Academic Editor

PLOS ONE
---

## [Editor Report · Acceptance letter]

15 Nov 2022

PONE-D-22-14963R2 

Metabolite profiling, antifungal, biofilm formation prevention and disruption of mature biofilm activities of *Erythrina senegalensis* stem bark extract against *Candida albicans* and *Candida glabrata*

Dear Dr. Harley:

I'm pleased to inform you that your manuscript has been deemed suitable for publication in PLOS ONE. Congratulations! Your manuscript is now with our production department. 

Kind regards, 

on behalf of

Professor Umakanta Sarker 

Academic Editor

PLOS ONE